# Non-Functional Jaw Muscular Activity in Patients with Disorders of Consciousness Revealed by A Long-Lasting Polygraphy

**DOI:** 10.3390/diagnostics13061053

**Published:** 2023-03-10

**Authors:** Martina Cacciatore, Francesca Giulia Magnani, Camilla Ippoliti, Filippo Barbadoro, Paola Anversa, Lara Portincaso, Elisa Visani, Jorge Navarro, Guya Devalle, Maurizio Lanfranchi, Valeria Pingue, Sara Marelli, Luigi Ferini Strambi, Francesca Lunardini, Simona Ferrante, Marco Tremolati, Matilde Leonardi, Davide Rossi Sebastiano, Davide Sattin

**Affiliations:** 1SC Neurologia, Salute Pubblica, Disabilità—Fondazione IRCCS Istituto Neurologico Carlo Besta, 20133 Milan, Italy; martina.cacciatore@istituto-besta.it (M.C.);; 2SC Neurofisiopatologia—Fondazione IRCCS Istituto Neurologico Carlo Besta, 20133 Milan, Italy; 3Istituti Clinici Zucchi, 20841 Carate Brianza, Italy; 4SC Epilettologia Clinica e Sperimentale—Fondazione IRCCS Istituto Neurologico Carlo Besta, 20133 Milan, Italy; 5IRCCS Fondazione Don Carlo Gnocchi, 20148 Milan, Italy; 6Vegetative State Unit—IRCCS Don Gnocchi Foundation, 20149 Milan, Italy; 7Rehabilitation Unit—Villa Beretta, Valduce Hospital, 23845 Lecco, Italy; 8Neurorehabilitation and Spinal Unit of Pavia Institute—Istituti Scientifici Maugeri IRCCS, 27100 Pavia, Italy; 9Department of Clinical Neurosciences, Neurology-Sleep Disorders Centre—IRCCS San Raffaele Scientific Institute, 20132 Milan, Italy; 10Sleep Disorders Center, Division of Neuroscience, Vita-Salute” San Raffaele University, 2013 Milan, Italy; 11Dipartimento di Neuroscienze Pediatriche—Fondazione IRCCS Istituto Neurologico Carlo Besta, 20133 Milan, Italy; 12Department of Electronics, Information and Bioengineering, Politecnico di Milano, 20133 Milan, Italy; 13Department of Biomedical, Surgical and Dental Sciences, University of Milan, 20122 Milan, Italy; 14Institute of Geriatric Rehabilitation, Pio Albergo Trivulzio, 20146 Milan, Italy; 15Istituti Clinici Scientifici Maugeri IRCCS, 20138 Milan, Italy; davide.sattin@icsmaugeri.it

**Keywords:** bruxism, myorhythmia, disorder of consciousness, unresponsive wakefulness syndrome, vegetative state, minimally conscious state

## Abstract

The presence of involuntary, non-functional jaw muscle activity (NFJMA) has not yet been assessed in patients with disorders of consciousness (DOC), although the presence of bruxism and other forms of movement disorders involving facial muscles is probably more frequent than believed. In this work, we evaluated twenty-two prolonged or chronic DOC patients with a long-lasting polygraphic recording to verify NFJMA occurrence and assess its neurophysiological patterns in this group of patients. A total of 5 out of 22 patients showed the presence of significant NFJMA with electromyographic patterns similar to what can be observed in non-DOC patients with bruxism, thus suggesting a disinhibition of masticatory motor nuclei from the cortical control. On the other hand, in two DOC patients, electromyographic patterns advised for the presence of myorhythmia, thus suggesting a brainstem/diencephalic involvement. Functional, non-invasive tools such as long-lasting polygraphic recordings should be extended to a larger sample of patients, since they are increasingly important in revealing disorders potentially severe and impacting the quality of life of DOC patients.

## 1. Introduction

Bruxism refers to a broad spectrum of non-functional, involuntary, repetitive jaw-muscle contraction resulting in phasic or tonic teeth gnashing, grinding, or clenching, usually associated with bracing or thrusting of the mandible [1]. Bruxism is a widespread disorder, with an undefined but certainly high prevalence in the general population, ranging from 8.0% to 31.4% (from 9.7% to 15.9% for sleep bruxism), depending on the different diagnostic methods used [2]. Complications of bruxism include dental wear or loss of teeth, tongue laceration or partial amputation, temporomandibular disorders, facial pain, and headaches [1,2].

From a pathophysiological point of view, the loss of cortical control on the trigeminal motor neuron nuclei probably plays a pivotal role in bruxism [3,4]. Additionally, cerebellar dysfunctions and proprioceptive trigeminal abnormalities [5], as well as anxiety, stress, and emotional factors, can favor it and exacerbate the symptoms [1,2,5].

In this framework, the lack of a large literature focused on bruxism in a coma and in the Disorders of Consciousness (DOC) is quite surprising: bruxism was first described in twenty comatose patients in 1985 by Pratap-Chand and Gourie-Devi [6]. Since then, only three works have been reported in the literature on the topic [7,8,9], and two of them are case reports. In critically ill patients, severe bruxism may enhance the difficulty in oral hygiene procedures, thus promoting a higher risk of infections and aspiration pneumonia [3,10]. Furthermore, bruxism, by determining arousals and phase changes, can worsen the duration and quality of sleep in DOC patients, thus hindering the recovery of a physiological sleep–wake cycle and resulting in lower residual cognitive levels [11]. Hence, it seems necessary to systematically verify the presence of bruxism in DOC patients, even considering the high rate of sleep disturbances affecting a similar population of patients [12].

Probably, one of the most relevant problems in investigating bruxism in DOC patients is that the non-instrumental diagnostic approaches [2] are limited to the clinical examination, while self-report of the symptoms is impossible. Thus, an approach based on a polygraphic recording (PG) is more suitable for detecting and evaluating bruxism in DOC patients. PG is low-cost, easy to apply at the patient’s bedside, repeatable, and free of side effects. In the last decade, neurophysiological techniques such as PG are growing in importance both in the diagnostic workup of DOC patients [11,13,14] as well as in the diagnosis of bruxism [1,15,16,17].

In this work, we evaluated twenty-two prolonged or chronic DOC patients with a simple, albeit long-lasting, PG recording at the patient’s bedside to characterize neurophysiological patterns of non-functional jaw muscle activity in this population. Furthermore, we compared the clinical profiles of two groups of patients, i.e., DOC patients with or without non-functional jaw muscle activity (NFJMA), to delineate different profiles between them, if any.

## 2. Materials and Methods

### 2.1. Patient Population, Diagnosis, and Behavioral Assessment

Twenty-two patients (11 females and 11 males, mean age 52.2 + 14.5 years) with prolonged or chronic DOC were consecutively enrolled from 1 June 2019 to 20 December 2022 as part of a national, multicentric clinical trial (EudraCT Number: 2019-001898-87) aimed at evaluating the tolerance and the efficacy of treatments for sleep disorders in DOC patients. This study was conducted following the Declaration of Helsinki, and it received approval from the ethics committee of the Fondazione-IRCCS-Istituto Neurologico “Carlo Besta” of Milan (approval number 51/2018). The written informed consent was obtained by patients’ legal representatives before the enrolment in this study.

The patients suffered from DOC after acquiring severe brain injury from different etiologies, with time from injury ranging from 2.6 to 107.5 months (mean value 21.7 months, median value 5.1), and they were hospitalized in rehabilitative or in long-term care structures; all of them were classified as Vegetative State/Unresponsive Wakefulness Syndrome (VS/UWS), Minimally Conscious State (MCS)-, and MCS+ and emerging from MCS (eMCS) on the basis of clinical and behavioral assessment. All the patients took a combination of sedative, antiepileptic, and cardiological drugs for their clinical needs, and they underwent daily rehabilitative treatment to prevent complications and reduce motor disability.

From two to five days after the enrolment, all the patients underwent an assessment, including the Italian version [18] of the Coma Recovery Scale-revised (CRS-r) [19]. CRS-r was administered three times, on three different days at different hours, by expert neuropsychologists (FM, MC, CI, and FB).

For each patient, age, times from injury, diagnosis, and the total and subscales CRS-r scores obtained at the best performances were collected for the subsequent analysis.

### 2.2. Long-Lasting Polygraphy and the Scoring of Non-Functional Jaw Muscle Activity

From seven to ten days after the enrolment (and three to five days after the last CRS-r evaluation), all the DOC patients underwent a 12 + 2 h PG, including night. PG was performed at the patient’s bedside, usually starting in the late evening and finishing in the following morning; the recording duration was limited during the day to prevent interferences with the daily clinical and rehabilitative activities.

All the recordings included 4–10 (depending on the integrity of the skull) electro-encephalographic (EEG) channels placed following the 10–20 International System, one or two electro-oculographic channels (depending on the integrity of the skull and the facial bones), four electromyographic (EMG) channels recorded from the mylohyoid (MYLO), masseter (MAS), and anterior tibialis muscles, a bipolar precordial electrocardiography, and an impedance-based thoracic pneumography, in patients without tracheostomy. The recordings were made using Ag/AgCl surface electrodes (impedance was kept below 5 kΩ), and the signals were acquired at a sampling rate of 512 Hz using a portable, multimode amplifier, BE-Micro (EBNeuro, Florence, Italy).

All the PG recordings were visually inspected by a clinical neurophysiologist (DRS) and a technician (PA) expert in the field of EEG in DOC patients, to determine the period of wakefulness and/or sleep on the basis of EEG and PG pattern(s) as well as the occurrence of significant NFJMA.

According to the recommendations proposed by the American Academy of Sleep Medicine for scoring bruxism [20], we established the following rules to define a single NFJMA:(1)Sustained (tonic) elevations of MASS and/or MYLO EMG activity that are at least twice the amplitude of the background activity for more than 2 s;(2)A sequence of at least three briefs (phasic) elevations of MASS and/or MYLO EMG activity that are at least twice the amplitude of the background activity for 0.25–2 s with a regular or pseudo-regular periodism.

In partial agreement with what has been previously performed in two works focused on bruxism in different patient populations [15,16], we considered a DOC patient with significant NFJMA if she/he showed during PG at least 4 episodes of phasic, tonic, and/or mixed NFJMA per hour in the awake or the sleep period, or both.

For the subsequent statistical analysis, DOC patients were divided into two groups on the basis of the PG, i.e., Group A and Group B, composed of patients with or without significant NFJMA, respectively.

### 2.3. Statistical Analysis

The following individual measures, i.e., age, time from injury, diagnosis (MCS- and MCS+ are united in the same class, namely MCS), etiology, CRS-r total, and CRS-r subscales scores of each patient of Group A and Group B were compared using Mann–Whitney U (U-Mann) or chi-squared tests, depending on the characteristics of the variables; the significance level was set at 0.05; Bonferroni correction was applied where appropriate. The data were analyzed using SPSS software, version 14 (SPSS Inc., Chicago, IL, USA).

## 3. Results

### 3.1. Patient Population, Diagnosis, and Behavioural Assessment

Demographic and clinical data of all patients are summarized in Table 1.

Immediately following the neuropsychological assessment and the PG recording, a total of 12, 7, and 2 patients were classified as VS/UWS, MCS-, and MCS+, respectively, while one patient was classified as eMCS. The same proportion of VS/UWS and MCS was found in both Group A and Group B (that is to say, in patients with or without NFJMA), i.e., 4/3 for Group A and 8/6 for Group B, even if the lack of MCS+ patients in Group A have to be reported.

From an etiological point of view, six, two, nine, three, and two patients suffered from traumatic, ischemic, hemorrhagic, anoxic, or mixed brain injury, respectively, with a similar distribution in the two groups, except for the hemorrhagic noxa that was prevalent in Group B with respect to Group A (8/15 patients v 1/7 patient).

The best scores obtained at the CRS-r and related sub-scales are shown in Table 2.

As expected, a huge variability across patients was found for total and sub-scale scores of the CRS-r; a comparison between the two groups showed very similar mean scores for the Visual, Oromotor, Communication, and Arousal sub-scales, while patients in Group B had slightly better total scores in the Auditory and Motor subscales than patients in Group A.

### 3.2. Long-Lasting Polygraphy and the Scoring of Non-Functional Jaw Muscle Activity

PG data revealed the presence of significant NFJMA in 7 of 22 patients, and during periods of awake, sleep, or both in 3, 1, and 3 patients, respectively. Similar to what was found in otherwise healthy patients with bruxism, we observed two forms of NFJMA in our patients:

(1) A sudden and sustained (tonic) contraction of MASS and MYLO becoming progressively weaker until returning to previous muscle tone in 2–10 s (see Figure 1A);

(2) Brief and phasic contractions of MASS and/or MYLO, recognizable on the corresponding traces as a biphasic, symmetric EMG activity waxes and wanes, typically in 0.25–0.5 s (only occasionally in 0.5–1 s) and usually repeated at regular or nearly regular intervals of 0.5–1.5 s and often limited to 4–15 events (see Figure 1B). In patients #1 and #5, NFJMA was continuous or subcontinuous for a very long time (in patient #1, up to 1 h and 20 min), always maintaining 0.75–1.5 Hz frequency.

Phasic and tonic NFJMA were observed in four patients and one patient, respectively, while in two patients, they both were present (Figure 1C). From an inspective point of view, NFJMA often polluted other channels, especially EEG channels placed over frontotemporal regions.

The range of frequencies of NFJMA varied a lot between patients, from 4.2 events/hour for patient #7 to the impressive 31.5 events/hours for patient #1 (mean frequency 9.6 + 9.9).

Results of the polygraphy and features of the NFJMA are shown in Table 3.

### 3.3. Statistical Analysis

No significant differences between Group A and Group B were found for age, sex, diagnosis, and time from injury. Hemorrhagic brain injury was significantly prevalent in Group B (*p* < 0.01). Again, no significant differences in total CRS-r and subscales scores were found, even if there was a trend for the CRS-r Motor subscale, whose scores are lower in the patients of Group A than those of Group B. Statistical analysis related to the CRS-r total and subscales scores are shown in Table 2.

## 4. Discussion

In this study, we aimed to explore the presence of NFJMA in a sample of DOC patients with a long-lasting PG; furthermore, we wanted to verify the differences in the residual cognitive functioning assessed by means of CRS-r scores in two groups of DOC patients defined by the presence or absence of NFJMA.

The first work on bruxism and its significance in the coma is dated 1985 [6]. In this pioneering work, the authors underlined the relationship between consciousness and bruxism, albeit without definitively clarifying it. In their sample, bruxism appeared with the return of a sleep–wake cycle and tended to disappear when a significant improvement in the level of consciousness was achieved [6]. Although interesting, this topic has not been further explored, and only a few case reports focused on NFJMA in DOC patients [7,8,9]. To date, no one has either described case series derived from a consecutive enrolment, or provided a precise description of how to diagnose bruxism in this specific and challenging context. This lack of interest could be explained by the compelling issues related to the diagnosis, clinical management, and rehabilitative needs of DOC patients, which, in part, can overshadow this intriguing topic that deserves more attention. Another reason why bruxism in DOC patients has been neglected is probably that self-report of the symptoms by patients is not possible [2,17]. As a result, disorders related to NFJMA presence are underrated in DOC patients.

Using a long-lasting PG, we demonstrated that 30.4% (7 out of 23) of enrolled DOC patients showed a significant presence of involuntary contractions involving the masticatory muscles. We do not consider representative the small sample of patients; hence it is not possible to determine a prevalence estimate of NFJMA in DOC patients. As a whole, the results are not entirely unpredictable, since bruxism or other hyperkinetic movement disorders involving the facial region (e.g., dyskinesia, dystonia, or even myoclonus) occurred in a wide range of acute brain-injured patients [3,21]. Moreover, a prevalence of about 5% for awake bruxism and 16.5% for sleep bruxism was found in a recent survey [2]. Nevertheless, it is the first time since the first work on bruxism and comas [6] that a study shows a similar prevalence of bruxism in patients with prolonged or chronic DOC. As a whole, these data underline the importance of systematic investigations of NFJMA and bruxism in DOC patients. Even considering all the difficulties associated with this severe condition, hypothesizing an effective treatment would be of the utmost importance to prevent complications such as dental wear, tooth loss, orofacial pain, and temporomandibular disorders, and to reduce the risk of oral infections and aspiration pneumonia [3,10,22].

CRS-r scores and other demographic and clinical variables considered did not differ significantly between the two groups of patients with or without NFJMA, even if there was a trend for the CRS-r Motor subscale, whose scores are lower in the patients with NFJMA than those without NFJMA. Further studies with a larger sample of patients would be useful to demonstrate whether there is an association between the presence of NFJMA and decreased levels of residual global functioning, some domains explored by the CRS-r scale, or even in the level of consciousness.

### Bruxism or Something Different?

Throughout the work, we preferred to use the term “non-functional jaw muscular activity” (NFJMA) over “bruxism” due to the fact that some PG patterns appeared dissimilar to what was usually observed in otherwise healthy patients suffering from bruxism.

NFJMA in patients #2, #3, #4, #6, and #7 is not so far from the “classical” form of bruxism, with the patients suffering either from awake or sleep-related NFJMA rather than both, with the awake presentation being more common. In these patients, the excessive activation of the masticatory muscles, especially of MASS, is often self-limiting, with a relatively small frequency of occurrence at about 4–5 events/hour.

Even if not definitely (since we only recorded patients once for about half a day), the dynamics of nocturnal sleep of patients #3 and #6 seem to be preserved, despite the presence of NFMJA. Indeed, these patients slept for more than 5 and 7 h, respectively, while in a previous work on 85 chronic DOC patients [23], the mean total sleep time was about 2.5 and just a little more than 3.5 h/night in VS/UWS and MCS patients, respectively. These data lead us to believe that the NFJMA of these patients is actually something very similar to bruxism [20]. Although much is unknown about the pathophysiology of bruxism, it is thought that temporary or enduring loss of the cortical control (and especially of the frontal lobes) decreases inhibition of the trigeminal motor neuron nuclei located in the pons, exiting in bruxism [3,4]. This pathophysiological mechanism could be enhanced in DOC patients due to the impairment of cortical control following brain injury.

On the other hand, patients #1 and #5 showed refractory, continuous, or sub-continuous, slow-frequency contractions of the masticatory muscles enduring for a long time, preventing the deepening of sleep and disrupting the normal dynamics of the sleep–wake cycle. This NFJMA looks like myorhythmia, a repetitive, rhythmic, and slow (ranging 1–4 Hz) movement disorder affecting mostly cranial and limb muscles [24]. Myorhythmia is an infrequent movement disorder usually related to Whipple’s disease. Myorhythmia may also occur in cerebrovascular disease, listeria encephalitis, anti-N-methyl-d-aspartate receptor encephalitis, encephalopathy associated with autoimmune thyroiditis, multiple sclerosis, and other disorders [24], and it is invariably associated with lesions involving the brainstem or diencephalic region, thus suggesting an involvement at these levels in patients #1 and #5.

## 5. Conclusions

Although much is unknown about the pathophysiology of the non-functional jaw muscular activity, the presence of bruxism and other forms of movement disorders involving facial muscles is probably an emerging problem more important than believed in DOC patients. Future studies with a larger sample of patients can contribute to clarifying the correlation between the presence of non-functional jaw muscular activity and the residual cognitive and motor performance, or the level of consciousness. Finally, a more accurate identification of specific polygraphic patterns of the orofacial movement disorders affecting DOC patients may lead to the identification of the involvement of different anatomical structures of the brain. Functional, non-invasive tools such as long-lasting PG should be extended to a larger sample of patients, since they are increasingly important in revealing disorders potentially severe and, in any case, with a great impact on the quality of life and the clinical management of the patients. Again, an advanced diagnostic workup of DOC patients should involve dentists and specialists in odontostomatology who can help clinicians in the management of orofacial disorders.

## Figures and Tables

**Figure 1 diagnostics-13-01053-f001:**
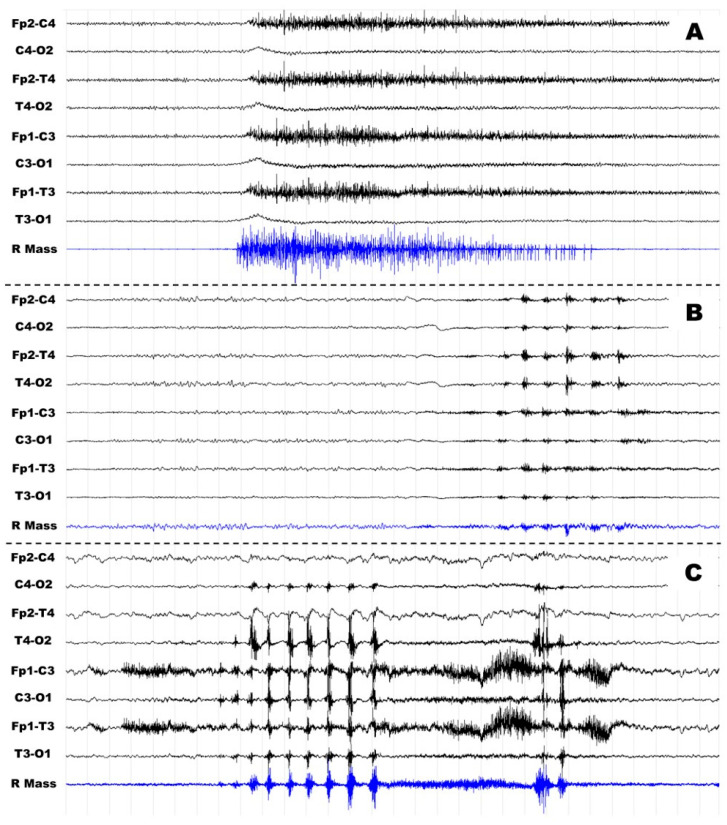
Examples of tonic, phasic, and mixed non-functional jaw muscle activity. Legend: example of tonic ((**A**), patient #4), phasic ((**B**), patient #7), and mixed ((**C**), patient #1) non-functional muscle activity. On each page, only EEG (black) and right-masseter EMG (blue) traces are shown. Time window 30 s, sampling rate 512 Hz, filters 1–70 Hz, gain 20 microV/mm.

**Table 1 diagnostics-13-01053-t001:** Demographic and clinical data of the sample.

Patient	Age (Years)	Sex	Etiology	Time from Injury (Months)	Disorder of Consciousness
Group A: DOC patients with non-functional jaw muscle activity					
#1	57.0	F	Mixed	26.6	VS/UWS
#2	24.2	F	Traumatic	3.9	MCS-
#3	60.9	M	Anoxic	57.4	VS/UWS
#4	43.9	F	Anoxic	25.0	VS/UWS
#5	57.1	M	Haemorrhagic	8.3	MCS-
#6	50.7	M	Traumatic	7.4	MCS-
#7	44.8	M	Traumatic	107.5	VS/UWS
Mean (SD)	48.3 (12.4)	NA	NA	33.7 (37.4)	NA
Group B: patients without non-functional jaw muscle activity					
#8	32.1	M	Traumatic	5.1	VS/UWS
#9	67.0	F	Haemorrhagic	4.5	VS/UWS
#10	75.1	F	Haemorrhagic	5.1	VS/UWS
#11	58.9	F	Haemorrhagic	71.9	VS/UWS
#12	49.4	F	Haemorrhagic	5.7	MCS-
#13	30.9	M	Mixed	103.7	VS/UWS
#14	38.3	F	Traumatic	5.4	VS/UWS
#15	59.8	F	Haemorrhagic	2.6	eMCS
#16	63.8	F	Haemorrhagic	5.0	MCS-
#17	68.0	M	Ischemic	12.9	MCS-
#18	27.5	F	Haemorrhagic	3.1	VS/UWS
#19	63.7	M	Haemorrhagic	1.7	VS/UWS
#20	70.3	M	Anoxic	4.5	MCS-
#21	54.1	M	Ischemic	4.2	MCS+
#22	51.7	M	Traumatic	4.9	MCS+
Mean (SD)	54.0 (15.4)	NA	NA	16.0 (29.9)	NA

Legend: VS/UWS: Vegetative State/Unresponsive Wakefulness State; MCS: Minimally Conscious State; eMCS: emerged from Minimally Conscious State; NA: not applicable.

**Table 2 diagnostics-13-01053-t002:** CRS-r scores of the patients.

Patient	CRS-R Scores						
	TOTAL	Auditory	Visual	Motor	Oromotor	Communication	Arousal
Group of patients with non-functional jaw muscle activity							
#1	3	1	0	0	0	0	2
#2	9	1	3	2	1	0	2
#3	2	0	0	1	1	0	0
#4	4	1	0	0	1	0	2
#5	9	2	2	2	1	0	2
#6	9	1	3	2	1	0	2
#7	4	1	0	0	1	0	2
Mean (SD)	5.7 (3.1)	1.0 (0.6)	1.1 (1.5)	1.0 (1.0)	0.9 (0.4)	0.0 (0.0)	1.7 (0.8)
Group of patients without non-functional jaw muscle activity							
#8	4	1	0	1	0	0	2
#9	4	1	1	0	0	0	2
#10	7	2	1	2	1	0	1
#11	3	0	0	1	1	0	1
#12	9	1	3	2	1	0	2
#13	5	1	1	0	1	0	2
#14	5	2	1	0	1	0	1
#15	20	4	5	6	2	1	2
#16	10	2	2	3	1	0	2
#17	16	2	4	5	3	0	2
#18	6	2	0	2	1	0	1
#19	2	0	0	1	0	0	1
#20	10	2	3	2	1	0	2
#21	12	4	0	5	1	0	2
#22	18	3	4	5	2	1	3
Mean (SD)	8.7 (5.6)	1.8 (1.2)	1.7 (1.7)	2.3 (2.0)	1.1 (0.8)	0.1 (0.4)	1.7 (0.6)
Comparison between groups (Mann–Whitney U test)							
*p*	0.164	0.166	0.310	0.083	0.640	0.286	0.958

**Table 3 diagnostics-13-01053-t003:** Results of the polygraphy and features of the non-functional jaw muscle activity.

Patient	Total Duration of the Recording (Hours)	Duration of Awake/Sleep Periods (Hours)	Number of Events	Frequency (Events/Hour)	Occurrence (Awake, Sleep, or Both)	Type (Phasic, Tonic, Mixed)
#1	13	11.4/1.6	409	31.5	both	Mixed (phasic predominant)
#2	12	9.3/2.7	72	5.8	both	Mixed
#3	12	6.8/5.2	62	5.2	awake	phasic
#4	14	11.1/2.9	68	4.9	awake	tonic
#5	12	10.8/1.2	133	11.1	both	phasic
#6	12	4.7/7.3	57	4.8	awake	phasic
#7	10	6.6/3.4	42	4.2	sleep	phasic
Mean (SD)	12.1 (1.2)	8.7 (2.6)/3.5 (2.1)	120.4 (130.5)	9.6 (9.9)	NA	NA

## Data Availability

Data will be available on request.

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
