# Peer review of "Non-Functional Jaw Muscular Activity in Patients with Disorders of Consciousness Revealed by A Long-Lasting Polygraphy"

_diagnostics, 2023, doi:10.3390/diagnostics13061053_

Round 1

Reviewer 1 Report

In this work, the authors were interested in the non-functional jaw muscular activity [NFJMA] in a population of twenty-two patients with disorders of consciousness [DOC] (12 UWS, 7 MCS-, 2 MCS+, 1 eMCS). The technique used was a 12-hour polygraphic recording. NFJMA was shown in 32% of patients awake or asleep for 3 out of 7, exclusively awake for 3 others or exclusively asleep for the last one. The PG recordings met the definition of bruxism for 5 of 7 NFJMA. This showed an increased prevalence of this phenomenon in the sample studied, not yet described.

The objective of the work is interesting and the method implemented to respond to it is simple and also compatible with clinical practice.This work complies with the regulations of good clinical research pratices (informed agreement of patient’s legal representative)

Minor comments

(A) Line 53 to line 79: Please remove these three unnecessary paragraphs from the work you present.

(B) Line 98: Replace “twenty-three” by “twenty-two”

(C) The incidence of bruxism in the general population varies from 5 to 21% according to the first article that described the phenomenon in the context of coma [Pratap-Chand and Gourie-Devi, Clin Neurol Neurosurg,1985]. A recent published survey concerning adult population (25-74 yo; 1200 subjects) find a prevalence of 5% for awake bruxism and 16.5% for sleep bruxism [Wetselaar et al. J Oral Rehabil, 2019]. In your small sample of patients with DOC, you find a prevalence of 22%.

Can you say that this prevalence differs from the general population?

Do you have estimation data using polygraphic recordings in general population for example?

Author Response

Milan, March 3rd, 2023

Comments and Suggestions for Authors
-reviewer#1

“Non-functional jaw muscular activity in disorders of con-sciousness patients revealed by a long-lasting polygraphy”

Diagnostics-2223264

In this work, the authors were interested in the non-functional jaw muscular activity [NFJMA] in a population of twenty-two patients with disorders of consciousness [DOC] (12 UWS, 7 MCS-, 2 MCS+, 1 eMCS). The technique used was a 12-hour polygraphic recording. NFJMA was shown in 32% of patients awake or asleep for 3 out of 7, exclusively awake for 3 others or exclusively asleep for the last one. The PG recordings met the definition of bruxism for 5 of 7 NFJMA. This showed an increased prevalence of this phenomenon in the sample studied, not yet described.

The objective of the work is interesting and the method implemented to respond to it is simple and also compatible with clinical practice. This work complies with the regulations of good clinical research pratices (informed agreement of patient’s legal representative)

Minor comments

(A) Line 53 to line 79: Please remove these three unnecessary paragraphs from the work you present.

(B) Line 98: Replace “twenty-three” by “twenty-two”

(C) The incidence of bruxism in the general population varies from 5 to 21% according to the first article that described the phenomenon in the context of coma [Pratap-Chand and Gourie-Devi, Clin Neurol Neurosurg,1985]. A recent published survey concerning adult population (25-74 yo; 1200 subjects) find a prevalence of 5% for awake bruxism and 16.5% for sleep bruxism [Wetselaar et al. J Oral Rehabil, 2019]. In your small sample of patients with DOC, you find a prevalence of 22%.

Can you say that this prevalence differs from the general population?

Do you have estimation data using polygraphic recordings in general population for example?

Reply to reviewer#1

First, we want to thank both the reviewers for their profitable suggestions and comments and even for their positive judgment of our work.

The point-by-point responses to Reviewer#1 are below (in italic font)

Line 53 to line 79: Please remove these three unnecessary paragraphs from the work you present.

After careful reading, we admit that the first paragraphs are completely unnecessary for the introduction of the topic of the work. We replaced them with a very short paragraph, and, at the same time, we enriched the Introduction by focusing on the main theme of the work, i.e. the non-functional jaw muscular activity.

Line 98: Replace “twenty-three” by “twenty-two”

We revised it in the text.

The incidence of bruxism in the general population varies from 5 to 21% according to the first article that described the phenomenon in the context of coma [Pratap-Chand and Gourie-Devi, Clin Neurol Neurosurg,1985]. A recent published survey concerning adult population (25-74 yo; 1200 subjects) find a prevalence of 5% for awake bruxism and 16.5% for sleep bruxism [Wetselaar et al. J Oral Rehabil, 2019]. In your small sample of patients with DOC, you find a prevalence of 22%.

Can you say that this prevalence differs from the general population?

Do you have estimation data using polygraphic recordings in general population for example?

Our small sample of subjects could not be considered representative of DOC patients, hence a generalization of our results in terms of the prevalence of bruxism would be incorrect. Again, we are not sure that bruxism may have a very different prevalence between the general population and chronic DOC patients. On the contrary, our work aims to underline how a relatively common problem such as bruxism (in the work cited by Reviewer #1, up to 16.5% of the general population) has so far been little or not at all considered in the literature in this class of patients, probably because the clinical and rehabilitative needs of them are so important as to overshadow everything else. Anyway, we added in the work the suggested survey and we modified a paragraph of the discussion to better clarify our point of view.  

Reviewer 2 Report

Dear Authors, 

The study is attractive and it is considered of scientific and even more clinical interest.

However, some suggestions should be made. 

The English language must be revised by a professional in order to improve grammar and vocabulary through the whole study. I recommend to avoid long sentences: short sentences are easier to understand.

The Abstract Section is well organized, and it clearly describes the research. 

The Introduction is well structured and explains the study rationale, but is very short and is needed to add other references to increase the quality of the manuscript, I suggest some recent references about TMD that can be useful: [doi: 10.1097/SCS.0000000000008771], [doi: 10.1097/SCS.0000000000009103];
[DOI: 10.1080/08869634.2022.2137129]

The Material and Methods section is adequate and well organized, but punctuation and spaces between words should be reviewed.

The Results and the Discussions are well structured, as well as the study limits. The weakness of this research is the lack of studies on this topic indeed, as the authors referred, future researches should be conducted with wider and multicentric samples.

The Conclusion briefly exposes the main findings of the study, underlying the clinical relevance of this study

The References should be better formatted as required by the Journal.

Finally, I suggest a Minor Revision for this Research.

Author Response

Milan, March 3rd, 2023

Comments and Suggestions for Authors
-reviewer#2

“Non-functional jaw muscular activity in disorders of con-sciousness patients revealed by a long-lasting polygraphy”

Diagnostics-2223264

Dear Authors, 

The study is attractive and it is considered of scientific and even more clinical interest.

However, some suggestions should be made. 

The English language must be revised by a professional in order to improve grammar and vocabulary through the whole study. I recommend to avoid long sentences: short sentences are easier to understand.

The Abstract Section is well organized, and it clearly describes the research. 

The Introduction is well structured and explains the study rationale, but is very short and is needed to add other references to increase the quality of the manuscript, I suggest some recent references about TMD that can be useful: [doi: 10.1097/SCS.0000000000008771], [doi: 10.1097/SCS.0000000000009103];

[DOI: 10.1080/08869634.2022.2137129]

The Material and Methods section is adequate and well organized, but punctuation and spaces between words should be reviewed.

The Results and the Discussions are well structured, as well as the study limits. The weakness of this research is the lack of studies on this topic indeed, as the authors referred, future researches should be conducted with wider and multicentric samples.

The Conclusion briefly exposes the main findings of the study, underlying the clinical relevance of this study

The References should be better formatted as required by the Journal.

Finally, I suggest a Minor Revision for this Research.

Reply to reviewer#2

First, we want to thank both the reviewers for their profitable suggestions and comments and even for their positive judgment of our work.

The point-by-point responses to Reviewer#2 are below (in italic font)

The English language must be revised by a professional in order to improve grammar and vocabulary through the whole study. I recommend to avoid long sentences: short sentences are easier to understand.

A Native-English speaker reviewed the language; moreover, for the sake of clarity, we modified long sentences and paragraphs as much as possible.

The Introduction is well structured and explains the study rationale, but is very short and is needed to add other references to increase the quality of the manuscript, I suggest some recent references about TMD that can be useful: [doi: 10.1097/SCS.0000000000008771], [doi: 10.1097/SCS.0000000000009103]; [DOI: 10.1080/08869634.2022.2137129].

Honestly, the Introduction is the paragraph that needed the most important changes. As suggested by reviewer#1, we removed from the manuscript the first paragraphs that are unnecessary and, at the same time, we tried to enrich the Introduction by focusing on the main theme of the work, i.e. the non-functional jaw muscular activity. In this line, we added two out of the three papers that you suggested. Despite being very interesting, the third one focused on the use of stem cells in the temporomandibular joint is further beyond the goal of our (preliminary) work, mainly focused on the diagnostic issues; hence I did not include it in the references.

The Material and Methods section is adequate and well organized, but punctuation and spaces between words should be reviewed.

We amended it in the text.

The Results and the Discussions are well structured, as well as the study limits. The weakness of this research is the lack of studies on this topic indeed, as the authors referred, future researches should be conducted with wider and multicentric samples. The Conclusion briefly exposes the main findings of the study, underlying the clinical relevance of this study

We have further expanded the discussion on the possible future research.

The References should be better formatted as required by the Journal.

We amended it in the references paragraph